# Humeral Head Three-Part Posterior Fracture-Dislocation Reduced through a Posterior Approach and Fixed with Blocked Threaded Wires: A Consecutive Case Series

**DOI:** 10.3390/medicina59040772

**Published:** 2023-04-16

**Authors:** Stefano Gumina, Vittorio Candela

**Affiliations:** Department of Anatomical, Histological, Forensic Medicine and Orthopaedics Sciences, Sapienza University of Rome, 00185 Rome, Italy; vittorio.candela@uniroma1.it

**Keywords:** posterior shoulder fracture dislocation, complex proximal humerus fracture dislocation, percutaneous blocked threaded wires, composite tuberosity shield fragment, posterior approach for posterior shoulder fracture-dislocation

## Abstract

*Background and Objectives:* Posterior fracture dislocations are rare. There is currently no uniformity regarding treatment. Therefore, outcomes are difficult to compare. We evaluated clinical and radiological outcomes of patients with humeral head posterior fracture dislocation treated with an open posterior reduction and then fixed with a biomechanically validated configuration of blocked threaded wires. *Materials and Methods*: 11 consecutive patients with humeral head three-part posterior fracture dislocation were treated by reduction through a posterior approach and fixed with blocked threaded wires. All patients were clinically and radiographically evaluated after a mean follow-up of 50 months. *Results:* The mean irCS was 86.1% (range: 70.5–95.3%). No significant difference was found between irCS at 6 and 12 months postoperatively and the final follow-up. Six patients noted their pain intensity as 0/10, three as 1/10, and two as 2/10. The postoperative reduction was considered as excellent in eight patients (Bahr’s criteria) and good in the remaining three; at the final follow-up, reduction was excellent and good in seven and four patients, respectively. The mean neck-shaft angles at FU 0 and at the final FU were 137° and 132°, respectively. No signs of avascular necrosis, non-union, and arthritis progression were seen. No recurrence of dislocation or posterior instability symptoms were reported. *Conclusions*: We believe that our very satisfactory results stem from: (1) the manual reduction of the dislocation through a vertical posterior surgical approach, which does not produce further osteocartilaginous damage of the humeral head; (2) no multiple perforations of the humeral head are performed; (3) the threaded wires have a smaller diameter than the screws, therefore they preserve the bone tissue of the humeral head; (4) deperiostization or further detachment of soft tissues are not expected; (5) the adopted and validated system is stable and limits translation, torsion, and the collapse of the humeral head.

## 1. Introduction

It is well known that posterior fracture dislocation of the humeral head is a rare event (0.6 per 100,000 population per year) [1]. In the Wronka et al. [2] series, consisting of 104 cases of fracture dislocation of the shoulder joint, only 10 cases were posterior (9.6%). Posterior fracture dislocation predominantly occurs in males aged between 35 to 55 years [1,3,4,5,6,7,8]; these injuries frequently occur as a consequence of seizure [4,5,9,10,11,12]; fall from height [1,4,7,9,13]; motor-vehicle accidents [4,5,7,13,14], or as a result of electrocution [4,15,16]. In a cohort of 164 patients with posterior fracture dislocation belonging to five different series [17,18,19,20,21], 24 (14%) had a bilateral posterior fracture dislocation.

Because of the rarity of this nosological entity and of different osteocartilaginous and capsololigamentous injuries that can be associated with posterior fracture dislocation, there is no uniformity of thought in the literature regarding how to manage it. In fact, a vast array of conservative or surgical treatments have been proposed: reduction of the dislocation and conservative treatment [2,5,22]; reduction and osteosuture [23]; reduction and anatomical reconstruction for reverse Hill Sachs lesion with bone graft [24,25]; classical or modified McLaughlin procedure [10,26,27,28,29]; reduction and synthesis with Kirschner/Steinmann wires [3,9,30,31] or with free screws [1,5,16,32]; reduction and synthesis with plate and screws [1,3,6,11,14,33,34,35,36]; humeral surface replacement [37]; hemiarthroplasty [2,6,10,15,31,33,38]; or arthroplasty [39].

Data relative to percutaneous pinning leads to further confusion. Using this system, many configurations can be obtained because wires and rods can be placed in different areas and directions according to the condition of the fracture, bone, and soft tissue as well as the surgeon’s experience. This variance generates confusion when interpreting results because of the poor reproducibility of the technique; often, the results depict the same fixation construct in all cases.

We aimed to evaluate the clinical and radiological outcomes of patients with humeral head three-part posterior fracture dislocation treated by reduction through a posterior approach and fixed with a biomechanically validated configuration [40,41] of blocked threaded wires, after a minimum follow-up of 24 months.

## 2. Materials and Methods

The study group was composed of 11 consecutive patients (9 M and 2 F; mean age (SD): 49.3 (7.2)), The diagnosis and classification of fractures were based on trauma series radiographs of the involved shoulder. Computed tomography scans were obtained in all cases in order to better evaluate the fracture pattern and to plan the surgery. In our series, we considered only patients who sustained a Neer 3-part posterior fracture dislocation. Eight patients had a Neer 3-part posterior fracture dislocation; in 3 patients, tuberosities formed a composite “shield” fragment, as described by Edelson et al. [42], through an intact periosteal sleeve.

Further exclusion criteria were chemotherapy, anticancer therapies in progress, sepsis, septic arthritis, osteomyelitis or other ongoing infectious processes, other systemic infectious processes, previous shoulder operations, patients receiving chronic therapy with steroids or nonsteroidal anti-inflammatory drugs, and patients with severe metabolic disorders.

Varus/valgus angulation of the upper humerus were assessed, as described by Majed et al. [43].

### 2.1. Surgical Technique

The patient was placed in beach chair position with the image intensifier on the same side as the fracture. All patients received “short antibiotic therapy,” which consists of 2 g cefazolin 30 min before surgery and 8 h and 16 h after surgery.

The medial edge of the scapula rested on the operating table. The entire upper limb was sterilized with chlorhexidine and betadine. Surgical drapes were positioned leaving the anterior and posterior aspect of the shoulder completely uncovered.

The surgical incision started posteriorly to the acromioclavicular joint and distally and longitudinally extended for about 8 cm. The deltoid fibers were bluntly separated. An auto-static retractor was then positioned to expose the muscular infraspinatus raphe. Just distally to the raphe, we performed a blunt transverse dissection which should have coincided with the cleavage plane between the infraspinatus and teres minor muscle. The access was maintained using a Gelpy retractor. In all cases, the posterior capsule was injured and the articular surface of the dislocated humeral head was facing posteriorly (Figure 1). Cautiously, in order not to damage the articular surface and not to cause further damage to the periarticular soft tissues, we manually reduced the humeral head. The posterior capsule and the cleavage between infraspinatus and teres minor were sutured. Fracture fixation was obtained using the technique for complex humeral head fractures proposed by Gumina et al. [40]. The threaded wires (2.5-mm diameter, fully threaded at the terminal 70 mm) were introduced through the lateral cortical bone of the humeral shaft 2 to 3 cm distal to the surgical neck with a distal proximal direction to the humeral head. In the sagittal plane, the wires had divergent directions, similar to the directions of the humeral load peaks described by Bergmann et al. [44]. According to their study, the load peaks on the humeral head occur in a superomedial direction in the frontal plane and a super posterior direction in the sagittal plane within a very small range of direction. Initially, wires did not have to cross the surgical neck. The reduction of displaced fractures by manipulation was attempted at this time. When a satisfactory reduction was not obtained, the varus/valgus deviation and the ante/retro torsion of the humeral head were corrected using a blunt elevator introduced through the posterior approach. When the reduction was achieved, the 2 wires were inserted through the surgical neck into the humeral head. A distance of 5 mm between the tip of the most cranial wire and the cortex of the humeral was maintained to avoid intraoperative humeral head perforation and postoperative humeral head impaction [45].

Reduction of the fragments, when dislocated, was conducted with a threaded wire or with a hook introduced through a 3 cm superior deltopectoral approach used to grasp the cuff tendons and relocate them to the correct position. Fixation was achieved with 2 threaded wires introduced via a craniocaudal and lateromedial direction in order to transfix the uppermost medial cortex of the humeral diaphysis. Two additional threaded wires were introduced to fix the tuberosities to the humeral head. Two threaded wires were then introduced into the upper portion of the humeral diaphysis, 4 to 5 cm distally to the surgical neck. The 4 couples of threaded wires were connected by a single rod. The deltoid split was closed (Figure 2a,b and Figure 3). No cases needed the use of a bone graft to fill humeral bone defects.

### 2.2. Postoperative Management

The involved shoulder was immobilized with the arm in a neutral position. The wires were medicated weekly. Patients were asked to start wrist, hand, and finger exercises and flexion/extension of the elbow starting from the first day post-op. After 1 month, the sling was removed and passive shoulder girdle exercises and passive shoulder motion were allowed; actively assisted shoulder motion was started not exceeding 120° of flexion and abduction with the contribution of a physiotherapist. At the 45th day post-op, the blocked threaded wires (Galaxy System^TM^, San Carlos, CA, USA) were removed and patients started full active exercises.

All patients were managed by the same physiotherapist.

### 2.3. Clinical Evaluation

Clinical evaluation was performed after 2, 3, 6, 12, 24, and 36 months and, when possible, 48, 72, and 96 months, after surgery. The individual relative Constant Murley score (irCS) was calculated at the final follow-up. No patients had contralateral shoulder dysfunction. Pain intensity was measured on the visual analogue scale (VAS).

### 2.4. Radiographic Evaluation

Anteroposterior and axillary view radiographs were performed 15 and 45 days after surgery to check reduction maintenance according to Bahrs criteria [46] and initial fracture healing, allowing us to remove the system; furthermore, 3, 6, 9, 12, 18, and 24 months after surgery, and then once per year, a shoulder X-ray was performed.

### 2.5. Statistical Analysis

Statistics were performed using SOFA Statistics version 1.5.3 (Paton-Simpson & Associates Ltd., Auckland, New Zealand); this software was also used for calculations, and data were analyzed by a single researcher. Independent T-Student test was used to analyze differences between the two groups. Categorical variables were calculated using frequencies and proportions while continuous data were estimated by means, standard deviations, and ranges. Calculated *p* values were 2-sided; a *p*-value of less than 0.05 was considered as significant and the range of confidence interval (CI) was 95%, where appropriate.

## 3. Results

The study group was composed of 11 consecutive patients (eight M and three F; mean age (SD): 53.3 (7.2)). Seven cases were caused by high energy trauma (motor vehicle accidents); three by sport trauma; and one by electrocution. The right shoulder was involved in eight cases.

The mean surgical time was 97 min (range: 67–132) and the interval between trauma and surgery was 5 days (range 2–11). The mean follow-up was 50 months (range 36–82).

No major complication occurred; three cases of superficial infection were treated with 5 days of oral antibiotics (amoxicillin/clavulanic acid 1 g × 3 times/day) with complete resolution.

The mean individual relative Constant score (irCS) at the final follow up was 86.1% (range: 70.5–95.3%). No significant difference was found between irCS at 6 (*p* = 0.215) and 12 months (*p* = 0.135) postoperatively and at the final follow up (*p* = 0.164) in all patients (Figure 4 and Figure 5).

Six patients (54.5%) described their pain intensity as 0/10, three (27.3%) as 1/10, and two (18.2%) as 2/10 according to the VAS.

According to the Bahr’s criteria, the postoperative reduction was considered excellent in eight patients and good in the remaining three (*p* = 0.322); at the final follow-up, reduction was excellent and good in seven and four patients (*p* = 0.252), respectively. In particular, the mean neck-shaft angles at follow-up 0 and final were 137° and 132°, respectively (*p* = 0.184).

No signs of avascular necrosis, non-union, and arthritis progression were seen during the radiographic follow-up. No recurrence of dislocation or posterior instability symptoms were reported.

## 4. Discussion

The posterior fracture dislocation is a rare nosologic condition whose pathological entities make each case unique. Unfortunately, it is difficult to easily interpret the literature data relating to the results obtained with surgical treatment because the various series are each made up of few cases and the patients included in the studies have often different initial anatomopathological pictures. Furthermore, the uncertainty is amplified by the fact that there is no uniformity of thought regarding: (1) how to treat the posterior fracture dislocation and, consequently, which type of surgical approach has to be used (deltopectoral [3,16,23,24,25,26,28,29,30,32,36,37,47], deltoid splitting [35], superior subacromial [4], vertical posterior [4,13,31], horizontal posterior [39], combined anterior and posterior [4,11,14], axillary [9] approach); (2) how to relocate the humeral head into the glenoid fossa (manually or by means surgical instruments that could cause further cartilage damage); and (3) how to fix the fracture dislocation (sutures, plates, K wires, screws) when the hypothesis of a prosthesis implant is not considered.

Soliman and Koptan [3] surgically treated 22 patients with a posterior fracture dislocation; of these, only seven were fixed with Kirschner or Steinmann wires. Of the seven cases, one had an anatomical neck fracture. Martens et al. [31] treated their patient with a posterior fracture dislocation using an analogous technique. Kaar [30], Limbosch [9], and Ide [47] used the percutaneous pinning technique for treating their three patients; in all three cases, the fracture line involved the anatomical neck.

Considering these papers, it is not possible to extrapolate how the Kirschner/Steinmann wires were inserted (insertion site and direction). Furthermore, none of the employed constructs had been anchored to an external rod. Therefore, the existing constructs did not have the possibility of limiting the torsion and translation of the fragments, and it is plausible that wires were placed in different areas and directions according to the condition of the fracture, bone, and soft tissue and the surgeon’s experience. All this generates confusion regarding the interpretation of results because of the poor reproducibility of the system.

To make the sample uniform, we considered only humeral head three-part posterior fracture dislocation with composite tuberosity shield fragment. This nosological condition is rare. In the Robinson et al. series [1], there were seventeen posterior fracture dislocations with composite tuberosity shield fragments that occurred in fifteen patients. However, in thirteen of the seventeen shoulders, there were one or more vertical intertubercular fracture lines, resulting in a “shattered shield” configuration, which remained a functionally, but not purely, composite through the intact periosteal sleeve.

The primary feature of the system we adopted is that the four rods are coplanar; that is, they lie on the same plane, thus forming a rigid system. The system is anchored to the bone at three points, identifying a plane that is kept in a fixed position by the system [40]. Any rotation of the bone (inside the arm) would change this relative position: hence, no rotation is possible. The bone below the line of fracture might slide in relation to the humeral head. With the construct we used, translation is impossible due to the presence of two threaded wires inserted in the diaphysis; these keep the bone at a fixed distance from the external structure. Postoperative “sintering effect” (threaded wire perforation of the humeral head) was avoided because we respected the 5 mm safe distance between the humeral articular surface and wire tip [45].

Although posterior fracture dislocation causes extensive damage to the bone, ligaments, and capsule, literature data surprisingly indicate that surgically treated patients report a satisfactory functional result and a Constant score value ranging between 75 and 89, regardless of the type of surgical treatment [1,4,6,8,14,24,26,27]. Our mean irCS score was 86.1%. No significant difference was found between irCS at 6 and 12 months postoperatively and at the final follow-up in all patients. Since the irCS score only approaches the absolute value of the Constant score in young patients, our results (taken from patients whose average age was over 50 years) may overlap with those reported in the literature.

Re-dislocation, avascular necrosis of the humeral head, non-union, malunion, resorption of the tuberosities, and subacromial impingement are well-known complications after surgical reduction and fixation of the posterior fracture dislocation of the humeral head [1,4,32,33,35]. These complications strongly depend on the fracture dislocation pattern. Fortunately, none of these complications were observed in our series. It is assumed that their absence is because: (1) the manual reduction of the dislocation, through a vertical posterior surgical approach, does not produce further osteocartilaginous damage of the humeral head caused by levers and retractors; (2) the adopted fixation system does not cause multiple perforations of the humeral head (which it is expected if we use plate and screws), which could further compromise its vascularization; (3) the threaded wires have a smaller diameter than the screws, therefore they preserve the bone tissue of the humeral head; (4) deperiostization or further detachment of soft tissues are not envisaged; (5) this type of fixation is considered stable as it limits translation, torsion, and collapse of the humeral head. This is confirmed by the radiologic outcomes observed in our series; the excellent immediate postoperative reduction was maintained in both the early postoperative (until device removal) and the last follow-up.

Epilepsy is a well-known cause of shoulder fractures, dislocations, and fracture dislocations. In our series, none of the eleven patients had a posterior fracture dislocation of the humeral head following a seizure episode. Although the construct that we have adopted guarantees stability of the fragment, we believe it is not advisable in epileptic patients due to the possible consequences related to the penetration of the metal wires that could occur following a sudden and uncontrolled fall.

The present study has limitations that need to be addressed: it is a consecutive case series, and no control group is present; however, a detailed literature review regarding alternative treatments has been performed. Further studies are needed in order to confirm these previous satisfactory results.

## 5. Conclusions

The treatment of humeral head three-part posterior fracture-dislocations reduced through a posterior approach and synthesized with blocked threaded wires leads to very satisfactory clinical and radiological outcomes. The manual reduction of the dislocation through a vertical posterior surgical approach, without producing further osteocartilaginous damage of the humeral head; the absence of multiple perforations of the humeral head with bone stock preservation; the absence of deperiostization or further detachment of soft tissues; and the stability of the fixation system, which prevents translation, torsion, and the collapse of the humeral head are all advantages of the technique.

## Figures and Tables

**Figure 1 medicina-59-00772-f001:**
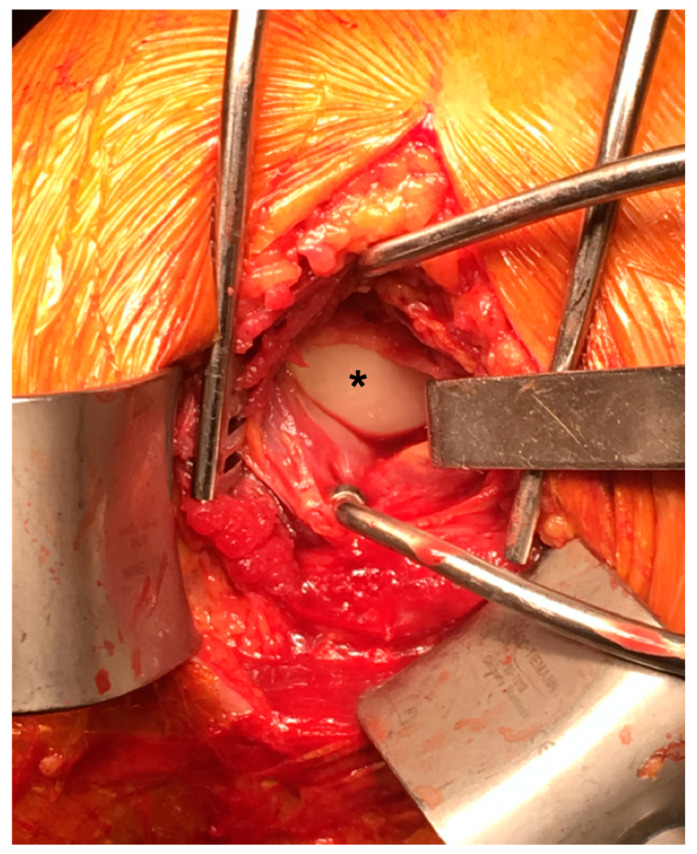
Left shoulder. Posterior Approach to the shoulder joint. The posterior capsule is injured, and the articular surface of the dislocated humeral head is facing posteriorly (*).

**Figure 2 medicina-59-00772-f002:**
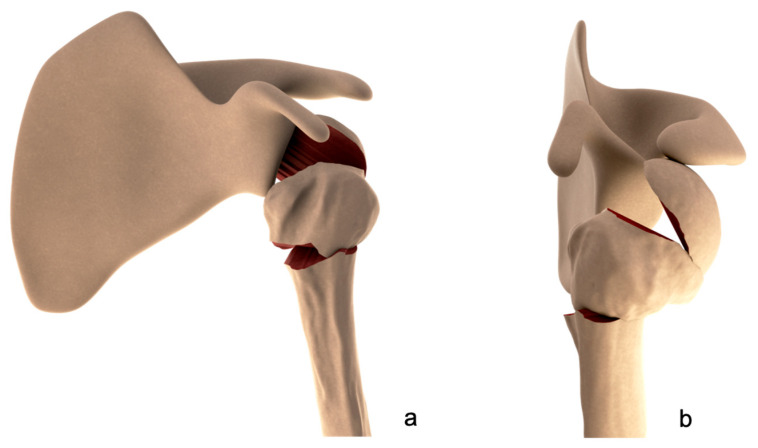
Drawings depicting a left three-part posterior fracture-dislocation of the humeral head with composite tuberosity shield fragment. Frontal (**a**) and lateral (**b**) views.

**Figure 3 medicina-59-00772-f003:**
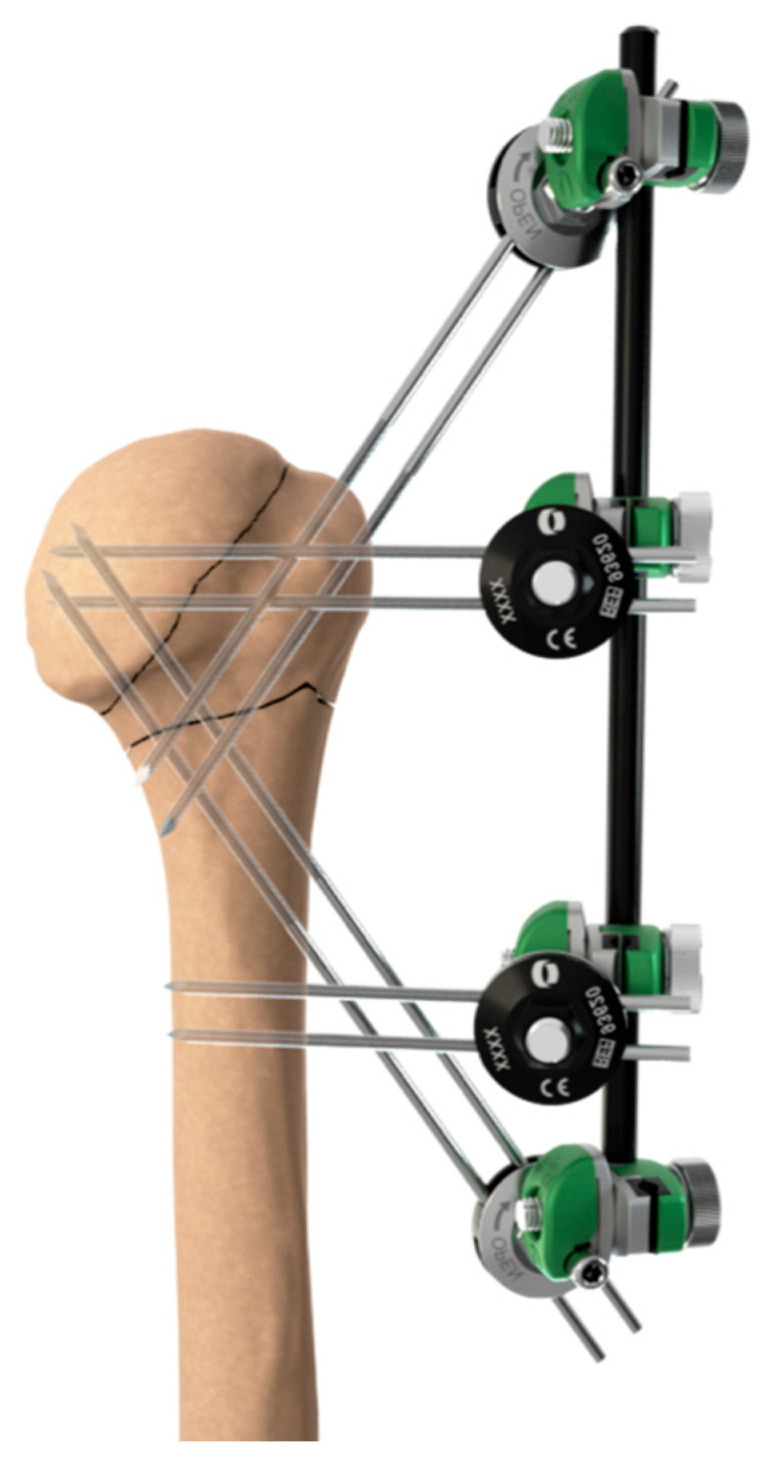
Left three-part posterior fracture-dislocation of the humeral head with composite tuberosity shield fragment treated with a biomechanically validated configuration of blocked threaded wires.

**Figure 4 medicina-59-00772-f004:**
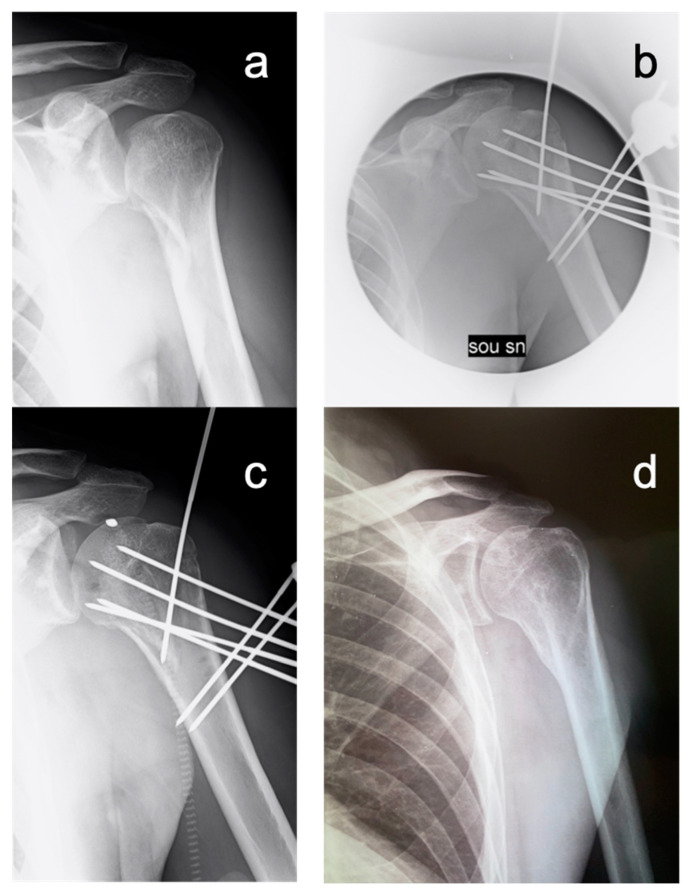
53ys male patient who sustained a left three-part posterior fracture-dislocation of the humeral head. (**a**) Preoperative X-ray; (**b**) Postoperative X-ray; (**c**) 45 day-FU X-ray; (**d**) 48 months FU X-ray.

**Figure 5 medicina-59-00772-f005:**
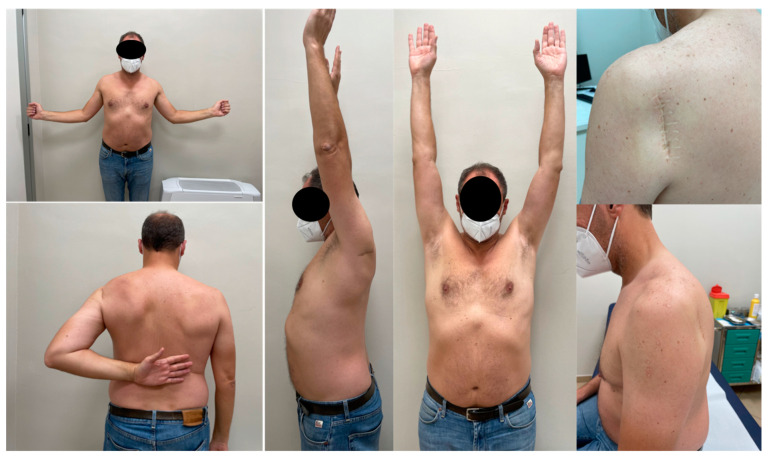
Final follow-up (48 months) and aesthetic results of a 53ys male patient who sustained a left three-part posterior fracture-dislocation of the humeral head with composite tuberosity shield fragment treated with a biomechanically validated configuration of blocked threaded wires.

## Data Availability

Not applicable.

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
