# Peer review of "Humeral Head Three-Part Posterior Fracture-Dislocation Reduced through a Posterior Approach and Fixed with Blocked Threaded Wires: A Consecutive Case Series"

_medicina, 2023, doi:10.3390/medicina59040772_

Round 1
Reviewer 1 Report
Well written and comprehensive report concerning a rare and perceived difficult-to-treat injury. Amazingly good results are reported, almost to good too be true. However, the arguments put forward that could explain the excellent results hold merit and are plausible.
The article should include a limitations section however, in which the limitations of a retrospective study without a control group are clearly stated. Obviously, the study could quite possibly be subject to a certain amount of selection bias and confirmation bias. Also, a sentence including the advice for at least an independent reaffirmation of the results should be included to alert the readers that the body of evidence has started with this valuable paper, however needs additional well conducted studies to conclude that this approach is indeed the way to go.
Author Response
Reviewer 1
The article should include a limitations section however, in which the limitations of a retrospective study without a control group are clearly stated. Obviously, the study could quite possibly be subject to a certain amount of selection bias and confirmation bias. Also, a sentence including the advice for at least an independent reaffirmation of the results should be included to alert the readers that the body of evidence has started with this valuable paper, however needs additional well conducted studies to conclude that this approach is indeed the way to go.
A limitation section has been added as suggested at the end of the discussion
The present study has limitations that need to be addressed: -it a consecutive case series; no control group is present; however, a detailed literature review regarding alternative treatments has been performed. Further studies are needed in order to confirm these previous satisfactory results.
Reviewer 2 Report
This is an interesting paper because of its focus on rare lesions.
I would like to point out some suggestions and comments:
# Results:
1) I suggest drawing a Table with all the patient's data and statistical methods that you have used. It would be very helpful to understand the results.
I couldn’t find any table.
2) I suggest to authors ( if possible) X-ray figures after fixation and Figures of patients using the external fixation.
Final comments/ Conclusion: I think this is an interesting study but I suggest adding some information to enhance the quality and help others surgeons to learn the technique.
Author Response
Reviewer 2:
# Results:
1) I suggest drawing a Table with all the patient's data and statistical methods that you have used. It would be very helpful to understand the results.
I couldn’t find any table.
2) I suggest to authors ( if possible) X-ray figures after fixation and Figures of patients using the external fixation.
A new table and a new figure have been added; figure 4 was renamed 5
(Table I and Figure 4)
Final comments/ Conclusion: I think this is an interesting study, but I suggest adding some information to enhance the quality and help other surgeons to learn the technique.
Two previous papers have been published in order to explain the surgical technique and to biomechanically validate it (reference 40-41).